# Combined Single Nucleotide Variants of *ORAI1* and *BLK* in a Child with Refractory Kawasaki Disease

**DOI:** 10.3390/children8060433

**Published:** 2021-05-21

**Authors:** Saki Kanda, Yoshimitsu Fujii, Shin-ichiro Hori, Taichi Ohmachi, Ken Yoshimura, Koichiro Higasa, Kazunari Kaneko

**Affiliations:** 1Department of Pediatrics, Kansai Medical University, Osaka, 2-5-1 Shin-machi, Hirakata-shi, Osaka 573-1010, Japan; kanda021513@gmail.com (S.K.); fujiiyos@hirakata.kmu.ac.jp (Y.F.); real14japan@yahoo.co.jp (S.-i.H.); omachit@hirakata.kmu.ac.jp (T.O.); yoshimura.kids.2525@gmail.com (K.Y.); 2Department of Genome Analysis, Institute of Biomedical Science, Kansai Medical University, Osaka 573-1010, Japan; higasako@hirakata.kmu.ac.jp

**Keywords:** *BLK*, cyclosporine A, *ORAI1*, refractory Kawasaki disease, single nucleotide variant

## Abstract

Kawasaki disease (KD) is a systemic vasculitis with an unknown etiology affecting young children. Although intravenous immunoglobulin (IVIG) plus acetylsalicylic acid is effective in most cases, approximately 10–20% of patients do not respond to this therapy. An 8-month-old boy was admitted to a local hospital with the presumptive diagnosis of KD. He received IVIG twice and four series of methylprednisolone pulse therapy from the third to the tenth day of illness. Despite these treatments, his fever persisted with the development of moderate dilatations of the coronary arteries. A diagnosis of refractory KD was made, and infliximab with oral prednisolone was administered without success. Defervescence was finally achieved by cyclosporine A, an inhibitor of the signaling pathway of the calcineurin/nuclear factor of activated T cells (NFAT). Whole-genome sequencing of his deoxyribonucleic acid samples disclosed two single nucleotide variants (SNVs) in disease-susceptibility genes in Japanese KD patients, *ORAI1* (rs3741596) and *BLK* (rs2254546). In summary, the refractory nature of the present case could be explained by the presence of combined SNVs in susceptibility genes associated with upregulation of the calcineurin/NFAT signaling pathway. It may provide insights for stratifying KD patients based on the SNVs in their susceptibility genes.

## 1. Introduction

Kawasaki disease (KD) is a systemic vasculitis affecting young children between 6 months and 4 years old. Although the therapeutic efficacy of intravenous immunoglobulin (IVIG) plus acetylsalicylic acid has been established for KD, approximately 10–20% of patients do not respond to IVIG [1]. As the patients resistant to IVIG have an increased risk of developing coronary artery abnormalities, additional therapeutic strategies such as IVIG retreatment, corticosteroids, or infliximab (IFX, an anti-tumor necrosis factor (TNF)-specific antibody) must be employed.

Recent studies have shown that some refractory KD cases had genetic backgrounds characterized by single nucleotide variants (SNVs) in disease-susceptibility genes [2,3,4].

Here, an infant with refractory KD who did not respond to repeated IVIG, corticosteroids, and IFX, but responded to cyclosporine A (CsA), an inhibitor of the calcineurin/nuclear factor of activated T cells (NFAT) signaling pathway, is presented. Whole-genome sequencing (WGS) revealed that he had combined SNVs in susceptibility genes reported in Japanese KD patients. We also discuss an association of the refractory nature of KD with SNVs.

## 2. Case Presentation

The patient’s clinical course during admission is shown in Figure 1 and Table 1. An 8-month-old boy presented with a 3-day history of persistent fever, conjunctival infection, changes in the oral mucosa and tongue, skin rash, and redness of the palms and the soles of the feet without identification of the causative infectious agents. A blood investigation showed leukocytosis and an increased level of serum C-reactive protein (CRP). Based on the presumptive diagnosis of KD, he received IVIG (2 g/kg/dose) twice and methylprednisolone pulse therapy (30 mg/kg/day) for four days at a local hospital between the third and the tenth day of illness, without defervescence. He was then transferred to us for further treatment on the tenth day of illness. On admission, the laboratory tests revealed leukocytosis and an elevated CRP level. An echocardiogram revealed moderate dilatations of the left anterior descending artery of the left coronary artery (LAD, Z-score 6.0) and right coronary artery (RCA, Z-score 8.2). The ethics committee approved this study of Kansai Medical University (no. 2019069).

From these findings, a diagnosis of refractory KD complicated by moderate coronary arterial lesions (CALs) was made, and IFX (5 mg/kg) with oral prednisolone (1 mg/kg/day) was administered intravenously on the 12th day of illness without success. Then, according to previous observations of efficacy for refractory KD [5], CsA was initiated orally (6 mg/kg/day) with a target trough level between 100–150 ng/mL from the 13th day of illness. In response to CsA, defervescence was observed on the 14th day of illness, and the dose of CsA was tapered off by two weeks. However, the re-development of fever with increased serum CRP soon after its cessation prompted us to recommence the CsA from the 30th day of illness, which controlled the disease activity well. After that, CsA was gradually tapered off by the 63rd day of illness. He developed CALs at the LAD (Z-score 21.8) and RCA (28.2).

At the outpatient clinic, at 1 year of age, echocardiographic examination demonstrated moderate CALs, necessitating the administration of warfarin, dipyridamole, and aspirin.

The patient’s parents provided informed consent and agreed with the submission of the case report to the journal.

## 3. Genetic Analysis

To determine the mechanism by which CsA was effective in this patient, WGS was performed to search for SNVs in the susceptibility genes in Japanese KD patients using NovaSeq6000 and the TruSeqDNA PCR-free library (Illumina Inc.; San Diego, CA, USA). The primary data from WGS were automatically processed for assembly and mapping. The obtained whole-genome data were subjected to structural annotation.

First, the susceptibility genes of Japanese KD were listed by a literature search as the following [2,3,4,6]: *ORAI1* (calcium release-activated calcium modulator), *FCGR2A* (Fc fragment of IgG, low-affinity IIa, receptor), *BLK* (B lymphoid tyrosine kinase), *CD40* (CD40), *ITPKC* (inositol 1,4,5-triphosphate 3-kinase C), *CASP3* (caspase-3), and *HLA* (human leukocyte antigen) (gene products were shown in parentheses). The present case had a total of 17,707 SNVs in these susceptibility genes. Among them, 17,692 SNVs were excluded from further analysis as they were not listed in Table 2, in which the reported SNVs associated with the susceptibility to KD were filled. As a result, 15 SNVs, including 4 SNVs in *ORAI1*, 4 in *CD40*, and 7 in *BLK,* were targeted for the next stage. After leaving out 12 SNVs of synonymous amino acid substitution, 3 SNVs, *ORAI1* (rs3741596), *CD40* (rs4813003), and *BLK* (rs2254546), remained as candidate SNVs associated with KD in the case. Among them, the SNV of *CD40* was determined to be neutral polymorphism based on the previous reports demonstrating that *CD40* (rs4813003) neither promoted nor inhibited the calcineurin/NFAT signaling pathway, preserving the function of CD40L-mediated signaling between B and T cells [7,8].

Thus, *ORAI1* (rs3741596) and *BLK* (rs2254546) were finally identified as significant SNVs in the case.

## 4. Discussion

Despite extensive research, the underlying mechanisms of KD remain unknown [1]. The current paradigm on this enigmatic vasculitis pathogenesis is that the disease results from an exaggerated immune response towards infectious agent(s) in a genetically and environmentally susceptible child [9,10]. Genetically determined susceptibility includes polymorphisms in the genes encoding cytokines, chemokines, and enzymes involved in signal transduction. Pathologically, KD is characterized by a marked immune activation associated with endothelial cell injury, which could be due to abnormal cytokine production. The clinical symptoms and laboratory findings also result from abnormally high inflammatory cytokines such as TNF-α, interleukin (IL)-1β, or IL-6 [11,12]. 

The therapeutic efficacy of IVIG has been established for KD, as IVIG reduces inflammation (fever, clinical signs, acute phase reactant levels) and prevents the development of CAL. IVIG appears to have a generalized anti-inflammatory effect, and possible mechanisms include the enhancement of regulatory T cell activity, the neutralization of bacterial super-antigens or other unknown pathogenic agents, the regulation of cytokine production, the suppression of antibody synthesis and inflammatory markers, the provision of anti-idiotypic antibodies, the Fc-gamma receptor and IL-1β, and balancing the T helper (Th) Th1/Th2 immune responses [13]. However, approximately 10–20% of patients do not respond to IVIG [1,11]. As IVIG-resistant patients have a higher probability for CAL formation [14,15,16], it is important to treat them aggressively. Mounting evidence has identified several epidemiological and laboratory characteristics as predictors of IVIG resistance [13]. These include age [16], illness day, platelet count, erythrocyte sedimentation rate, hemoglobin concentration, CRP [15,16], eosinophils, lactate dehydrogenase, albumin, alanine aminotransferase, concomitant infection [10,15], and granulocyte-colony stimulating factor [14]. Several treatments are available for combination with the second administration of IVIG for patients who do not respond to the initial IVIG treatment, such as IFX, corticosteroids, cytotoxic drugs (cyclophosphamide, methotrexate, CsA), plasmapheresis, and plasma exchange [13].

Though the pathogenesis is poorly understood, recent advances in genome-wide association studies (GWAS) and linkage analyses of KD have identified the SNVs in several disease-susceptibility genes. Candidate genes can be classified into four groups: enhanced T cell activation (*ITPKC, ORAI1, STIM1*), dysregulated leukocyte signaling (*CD40, BLK, FCGR2A*), decreased apoptosis (*CASP3*), and altered transforming growth factor-beta signaling (*TGFB2, TGFBR2, MMP, SMAD*) [3]. Among them, eight SNVs in seven genes are reported to be representative in Japanese KD patients. These genes (SNVs) are *ITPKC* (rs28493229), *ORAI1* (rs3741596, rs141919534), *FCGR2A* (rs1801274), *CD40* (rs4813003), *BLK* (rs2254546), *CASP3* (rs113420705), and *HLA* (rs2857151) [2,3,4,6]. Interestingly, most molecules encoded by these genes play important roles in the calcineurin/NFAT signaling pathway (Figure 2). Therefore, it is reasonable to speculate that SNVs in these susceptibility genes could also be related to the refractory nature in the present case via functional changes, with an upregulation of the signaling pathway. SNVs in *ITPKC* and *CASP3* are reported to be associated with resistance to IVIG [17]. Furthermore, recent observations have revealed that one SNV in *ORAI1* in a refractory patient caused a gain of function, leading to constitutive Ca^2+^ entry into immune cells, which upregulated the calcineurin/NFAT signaling pathway [18]. Therefore, the SNV in *ORAI1* (rs3741596) detected in the present case might be associated with its refractory nature. 

Meanwhile, another SNV in *BLK* (rs2254546) found in the case may also be associated with the activation of the calcineurin/NFAT signaling pathway via cross-interaction between T cells and B cells [19,20]. In this regard, Simpfendorfer et al. revealed that patients with rheumatoid arthritis having SNVs in *BLK*, including rs2254546, demonstrated a decreased expression of *BLK* on B cells with reduced B cell receptor signaling activity, leading to the activation of T cells via the CD86 molecule [21]. This report supports the concept that the SNV in *BLK* detected in the present case (rs2254546) could amplify the upregulation of the calcineurin/NFAT signaling pathway.

Recent observation confirmed the involvement of this pathway in KD, demonstrating that the serum levels of calcineurin and NFAT increased significantly in the acute stage and decreased progressively in the afebrile and subacute stage, reducing the CRP, white blood cell, and neutrophil counts [22]. Furthermore, an in vitro study using cultured endothelial cells of the coronary artery also suggested that the activation of the calcineurin/NFAT signaling pathway plays a vital role in causing the inflammation of the endothelial cells; Wang et al. demonstrated that the serum of KD patients promoted the proliferation of coronary artery endothelial cells not only morphologically but also by increasing the expression of proteins involved in cell proliferation and proteins constituting the calcineurin/NFAT signaling pathway. Furthermore, they reported that the expression of these proteins was suppressed by adding CsA to KD patients’ serum [23]. In other words, the calcineurin/NFAT signaling pathway activated by some extracellular inflammatory molecules is involved in CAL formation, and CsA was shown to be effective in preventing CAL formation in vitro.

From these findings, CsA, a calcineurin inhibitor, is expected to suppress the upregulation of the calcineurin/NFAT signaling pathway in refractory Japanese KD patients. The recent clinical trial from Japan revealed the efficacy of CsA in KD; an open-label, randomized, multicenter study of 173 patients with KD showed that IVIG plus CsA resulted in earlier fever resolution, lower CRP, and a lower CAL formation rate than IVIG alone. There was no difference in adverse events between the two groups, confirming that CsA is helpful in clinical practice for calming inflammation and preventing CAL formation in KD [24].

Based on these findings, the following mechanisms might be involved in the efficacy of CsA in the present case: the calcineurin/NFAT signaling pathway was constitutively upregulated because of an elevated cytosolic Ca^2+^ concentration due to functional changes in ORAI1 due to SNV in *ORAI1* (rs3741596); another SNV in *BLK* (rs2254546) might indirectly aggravate the calcineurin/NFAT signaling pathway, with an increase in T cell–B cell collaboration; as a result, upregulated calcineurin/NFAT signaling pathway caused an enhanced proinflammatory cytokine production leading to multidrug resistance including IVIG, corticosteroids, and IFX; CsA effectively suppressed the upregulated calcineurin/NFAT signaling pathway resulting in remission, which could not be obtained by other therapeutic agents including repeated IVIG or IFX.

Our present case study has two limitations. First, our concept concerning the relationship between multidrug resistance and the susceptibility genes in KD is based on only a single case. To verify this, further studies with more cases are necessary. Second, the genetic examination was performed using the targeted analysis for the annotation. Thus, several SNVs of other candidate genes have not been considered for their association with KD. 

## 5. Conclusions

The refractory nature of KD patients who did not respond to conventional therapies could be explained by the presence of combined SNVs in susceptibility genes associated with the upregulation of the calcineurin/NFAT signaling pathway. Our findings may provide insights for stratifying KD patients based on the SNVs in their susceptibility genes.

## Figures and Tables

**Figure 1 children-08-00433-f001:**
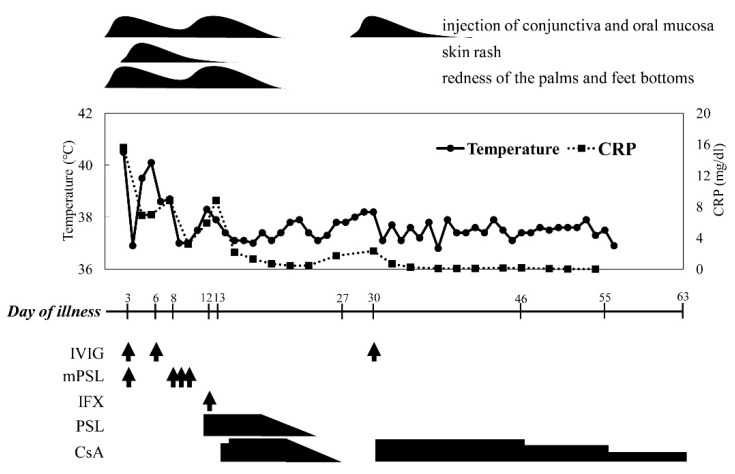
The clinical course of the present case. Abbreviations and administered dose of the therapeutic agents: CRP, C-reactive protein; IVIG, intravenous immunoglobulin (2 g/kg/day); m-PSL, methylprednisolone (30 mg/kg/day); IFX, infliximab (5 mg/kg/day); PSL, prednisolone (1 mg/kg/day per os); CsA, cyclosporine A (6 mg/kg/day per os with a target trough level between 100–150 ng/mL); arrows indicate the intravenous administrations of therapeutic agents.

**Figure 2 children-08-00433-f002:**
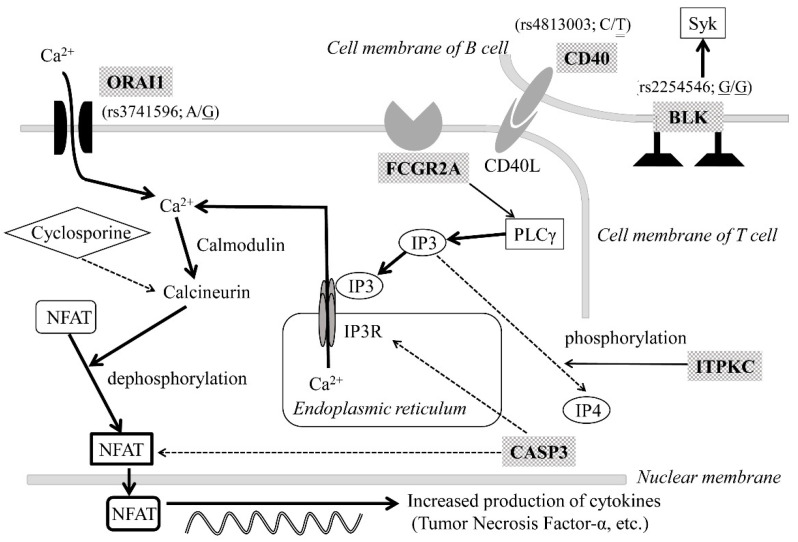
Susceptibility genes in Kawasaki disease in Japanese patients encoding molecules associated with T-cell activation. The present case had three significant single nucleotide variants (SNVs): *ORAI1* (rs3741596), *CD40* (rs4813003), or *BLK* (rs2254546). The types of SNVs, base substitutions, and genotypic patterns of the case are shown in parentheses. The single underline shows an alternative allele with risk for KD, and the double line means a neutral polymorphism. Shaded molecular names are encoded by the representative disease susceptibility genes in Japanese KD patients. Solid lines with arrows indicate activation, while dashed lines with arrows denote the suppression of the calcineurin/NFAT signaling pathway. Abbreviations: BLK, B lymphoid tyrosine kinase; CASP3, caspase-3; FCGR2A, Fc fragment of IgG low-affinity IIa receptor; He, heterozygous genotype; Ho, homozygous genotype; ITPKC, inositol 1, 4, 5-triphosphate 3 kinase C; IP3: inositol 1,4,5-triphosphate; IP3R, inositol 1,4,5-triphosphate receptor; KD: Kawasaki disease, NFAT, nuclear factor of activated T cells; ORAI1, calcium release-activated calcium modulator 1; PLC, phospholipase C; Syk, spleen tyrosine kinase.

**Table 1 children-08-00433-t001:** Laboratory and Echocardiogram Data in the Clinical Course.

	Day of Illness					
	3	10	12	14	30	84
White Blood Cell (/μL)	13,500	37,600	37,300	29,100	13,600	10,800
Neutrophil (/μL)	10,700	21,400	18,500	11,800	5000	5000
Platelet Cell (×109/L)	297	667	808	815	485	369
Albumin (g/L)	42	26	28	27	37	NA ^(g)^
CRP ^(a)^ (mg/L)	156.0	32.2	59.1	88.1	23.1	NA ^(g)^
NT-pro-BNP ^(b)^ (pg/mL)	4560.0	1406.5	633.5	514.7	328.1	NA ^(g)^
AST ^(c)^ (U/L)	174	29	27	26	30	NA ^(g)^
ALT ^(d)^ (U/L)	106	33	23	23	10	NA ^(g)^
Diameter of LAD ^(e)^ (mm)	1.5 [0.6]	2.9 [6.0]	3.6 [9.2]	4.8 [14.8]	6.3 [21.8]	6.0 [20.2]
Diameter of RCA ^(f)^ (mm)	1.5 [0.4]	4.0 [8.2]	4.0 [8.2]	5.6 [13.7]	9.9 [28.2]	8.5 [23.3]

Numbers in square brackets denote the coronary artery z-scores. Abbreviations: ^(a)^ CRP: C-reactive protein, ^(b)^ NT-pro-BNP: N-terminal pro-brain natriuretic peptide, ^(c)^ AST: aspartate aminotransferase, ^(d)^ ALT: alanine aminotransferase, ^(e)^ LAD: left anterior descending coronary artery, ^(f)^ RCA: right coronary artery, ^(g)^ NA: not available.

**Table 2 children-08-00433-t002:** Studied single-nucleotide variants in the candidate genes in Japanese Kawasaki disease.

*ORAI1*	*CD40*	*BLK*	*FCGR2A*	*CASP3*	*ITPKC*	*HLA*
rs868908978	rs4813003	rs2254546	rs10401344	rs113420705	rs28493229	rs2857151
rs141919534	rs4810485	rs2736340	rs17713068			
rs782675422	rs6074022	rs13277113	rs2233152			
rs543433737	rs3746821	rs2618476	es10403040			
rs781980977	rs1569723	rs12680762	rs1801274			
rs3741595	rs1883832	rs2736340				
rs782238081	rs1569723	rs6993775				
rs3741596	rs4813003	rs1382566				
rs782308800	rs199581355					
rs782722476						
rs377456337						
rs200214435						
rs3741597						
rs781789915						
rs3825174						
rs3825175						
rs555170508						
rs375464035						
rs12313273						
rs6486789						
rs117324670						
rs7484839						
rs7486943						
rs712853						

All studied single nucleotide variants were reported by the genome-wide association study of Kawasaki disease in Japan. [2,3,4,6]. Abbreviations: *ORAI1*, calcium release-activated calcium modulator 1; *BLK*, B lymphoid tyrosine kinase; *FCGR2A*, Fc fragment of IgG low-affinity IIa receptor; *CASP3*, caspase-3; *ITPKC*, inositol 1, 4, 5-triphosphate 3 kinase C; *HLA*, human leukocyte antigen; rs, reference of single nucleotide polymorphism.

## Data Availability

It is not applicable in this section.

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
