# Peer review of "Combined Single Nucleotide Variants of ORAI1 and BLK in a Child with Refractory Kawasaki Disease"

_children, 2021, doi:10.3390/children8060433_

Round 1

Reviewer 1 Report

The authors presented an 8-month-old male infant with refractory KD, who did not respond to repeated IVIG, corticosteroids, and infliximab (IFX) but responded to cyclosporine A (CsA). WGS study disclosed three SNVs, including ORAI1 (rs3741596), CD40 (rs4813003), and BLK (rs2254546) are associated with KD in the case. The authors concluded that KD patient refractory to conventional therapies could be explained by the presence of combined SNVs in susceptibility genes related to upregulation of the Ca2+/NFAT-pathway. This report describes uncommon conditions that serve to enhance our medical care in KD. I have only one issue I feel need to be addressed and would welcome the author’s comments on these.

1.      Since that ORAI1 is involved in T cell activation, and that CD40 and BLK are related to dysregulated B cell signaling, why this patient refractory to IFX but responding to CsA therapy? What is the role of CD40 SNV in the Ca2+/NFAT pathway?

        I hope that the enclosed comments will be of help to the authors.

Author Response

Please find the attached letter to Reviewer 1.

Reviewer 2 Report

The report by Kanda et al., entitled “Combined single nucleotide variants of ORAI1 and BLK in a child with refractory Kawasaki disease (KD)”, dealing with an 8-month-old boy hospitalized due to  multi-resistant KD, is interesting to read.

Infliximab, prednisolone and cyclosporine were necessary to control patient’s fever. Whole-genome sequencing revealed peculiar variants in disease-susceptibility genes of Japanese KD patients: ORAI1 (rs3741596) and BLK (rs2254546). The authors conclude that the refractoriness of this KD might be explained by the presence of those variants in KD susceptibility genes.

The limitation of this single report has been declared.

In the discussion you should try to give a wider breath to your considerations, describing which KD elements can predict responsiveness to IVIG and writing that vascular complications (and response to IVIG) for KD are related to the “extent “of overall inflammation, as shown by different papers (Dionne A, et al. PLoS One 2018, 13: e0206001; Rigante D, et al. Rheumatol Int 2010, 30: 841-6; Downie ML, et al. J Pediatr 2016, 179: 124-30; Lee CP, et al. Pediatr Res 2013, 74: 545-51; Abe J, et al. J Allergy Clin Immunol 2008, 122: 1008-13), which it is important to cite.

Please, clarify or simplify the sentence: “the Ca2+/NFAT pathway was constitutively upregulated by an elevated cytosolic Ca2+ concentration with enhanced cytokine production in the immune cells and that CsA effectively suppressed the Ca2+/NFAT pathway resulting in remission, which other therapeutic agents could not be obtained”.

Author Response

Please find the attached letter to Reviewer 2.
